# Match Point: Nuclear Medicine Imaging for Recurrent Thyroid Cancer in TENIS Syndrome—Systematic Review and Meta-Analysis

**DOI:** 10.3390/jcm13185362

**Published:** 2024-09-10

**Authors:** Fabrizia Gelardi, Alexandra Lazar, Gaia Ninatti, Cristiano Pini, Arturo Chiti, Markus Luster, Friederike Eilsberger, Martina Sollini

**Affiliations:** 1Faculty of Medicine, Vita-Salute San Raffaele University, 20132 Milano, Italy; 2IRCCS San Raffaele Hospital, 20132 Milano, Italy; 3School of Medicine and Surgery, University of Milano-Bicocca, 20900 Monza, Italy; 4Nuclear Medicine Department, University of Marburg, 35037 Marburg, Germany; 5IRCCS Humanitas Research Hospital, 20089 Rozzano, Italy

**Keywords:** thyroid cancer, TENIS syndrome, iodine-131, nuclear medicine, PET/CT, theranostics

## Abstract

**Background/Objectives:** Disease recurrence and resistance to radioiodine (RAI) therapy are major challenges in the management of differentiated thyroid cancer (DTC). In particular, the TENIS (Thyroglobulin Elevated Negative Iodine Scintigraphy) syndrome, characterised by elevated thyroglobulin (Tg) serum levels in addition to a negative radioiodine whole body scan (WBS), complicates disease monitoring and treatment decisions. Conventional imaging techniques often fail to detect disease in WBS-negative patients with rising Tg levels, leading to limitations in therapeutic intervention. This systematic review and meta-analysis aims to evaluate the diagnostic accuracy of nuclear imaging modalities in detecting disease recurrence in patients with the TENIS syndrome and to provide insights to guide therapeutic approaches in this complex clinical scenario. **Methods**: A comprehensive search of PubMed/MEDLINE and EMBASE databases up to March 2024 was performed according to PRISMA guidelines. Eligible studies were selected, and quality assessment was performed with the QUADAS-2 tool. For each study, relevant data were extracted and synthesised. A meta-analysis of the diagnostic accuracy of [^18^F]FDG PET/CT was performed, and patient-based pooled sensitivity and specificity were calculated using a random-effects model. Statistical heterogeneity between studies was assessed using the I2 statistic. **Results**: Of the 538 studies initially identified, 22 were included in the systematic review, of which 18 were eligible for meta-analysis. The eligible studies, mainly focused on [^18^F]FDG PET/CT, showed variable sensitivity and specificity for the detection of RAI-refractory thyroid cancer lesions. For [^18^F]FDG PET/CT, pooled estimates displayed a sensitivity of 0.87 (95% CI: 0.82–0.90) and a specificity of 0.76 (95% CI: 0.61–0.86), with moderate heterogeneity between studies. **Conclusions**: [^18^F]FDG PET/CT remains central in the detection of disease recurrence in patients with the TENIS syndrome. The emergence of novel radiopharmaceuticals with specific molecular targets is a promising way to overcome the limitations of [^18^F]FDG in these patients and to open new theranostics perspectives. This review highlights the great potential of nuclear medicine in guiding therapeutic strategies for RAI-refractory thyroid cancer.

## 1. Introduction

Differentiated thyroid cancer (DTC) presents a favourable prognosis, with a 5-year survival rate of over 98% [1]. Radioiodine whole body scan (WBS) is a pivotal diagnostic tool for the clinical management of DTC during various stages of the natural course of the disease [2,3]. About 10% of patients develop distant metastases during the course of the disease, and two-thirds of them become refractory to treatment with radioactive iodine (RAI) [4]. Metastatic DTC patients often experience rapidly progressive disease (with an average interval between the first and second metastases of <15 months [5]), and their prognosis further worsens when RAI refractoriness occurs (10-year survival rate < 10% with a mean life expectancy of 3–5 years) [6].

In this context, the TENIS syndrome, characterised by elevated thyroglobulin serum levels in addition to negative radioiodine WBS, represents a particularly complex clinical challenge. Several factors could explain this mismatch between serum and imaging biomarkers, including competitive inhibition of RAI uptake, tumour dedifferentiation, loss-of-function mutations in the sodium–iodide symporter, and lesions below the detection threshold size [7]. Conventional imaging techniques often fail in the early detection of disease recurrence in RAI-negative patients with rising Tg levels. As a result, the management of patients with the TENIS syndrome remains heavily dependent on monitoring Tg levels until visual detection of disease, with therapeutic intervention limited to systemic treatments of uncertain efficacy. The effectiveness of empirical radioiodine treatment in this scenario is still debated [8,9,10]. Alternative treatments, such as external beam radiotherapy, multikinase inhibitors, and redifferentiating agents, offer potential avenues in the field of RAI-refractory thyroid cancer [10,11]. In such a heterogeneous and complex scenario, nuclear medicine approaches can provide invaluable insights into tumour metabolism and can facilitate the identification and characterisation of thyroid lesions [12]. Furthermore, nuclear medicine represents a promising therapeutic frontier beyond RAI, with some studies suggesting benefits from peptide receptor radionuclide therapy (PRRT) and other radioligand therapies (RLTs) [13,14,15,16,17] when established treatments are no longer effective. Our systematic review and meta-analysis aim to evaluate the diagnostic accuracy of nuclear medicine imaging modalities in detecting disease recurrence in patients with the TENIS syndrome and in guiding therapeutic strategies in this challenging clinical setting.

## 2. Materials and Methods

This systematic review was conducted in accordance with the Preferred Reporting Items for Systematic Reviews and Meta-Analyses (PRISMA) guidelines [18] and the associated checklist. The review was not registered on an online database.

### 2.1. Literature Search Strategy

A comprehensive literature search was performed using PubMed/MEDLINE and EMBASE databases to identify original research studies relevant to the scope of the present review and meta-analysis. The search strategy employed multiple combinations of the following keywords: (“thyroid”) AND (“TENIS” OR “dedifferentiated” OR “WBS-negative” OR “iodine-negative” OR “131I-negative” OR “resistant”) AND (“scintigraphy” OR “SPECT” OR “PET/CT”). The search was completed on 3 March 2024.

### 2.2. Study Selection Process

The resulting lists of matching articles were exported in .csv format and merged into a single file. Titles and abstracts were first screened to remove duplicates and to exclude articles that were outside the scope of the present work. Exclusion criteria for this stage included the following: (1) Reviews, meta-analyses, letters, commentaries, editorials, guidelines, book chapters, and case reports; (2) Studies not involving humans; (3) Articles not related to the area of interest; (4) Articles without an available English translation. Selected articles underwent full-text screening to further refine the inclusion criteria. Exclusion criteria at this stage included the following: (5) Articles without an available full text; (6) Studies involving <15 patients; (7) Articles in which the interval between negative WBS and the imaging study exceeded 6 months or was not specified; (8) Studies involving patients positive on WBS not analysed separately; (9) Studies involving patients who did not undergo WBS.

### 2.3. Quality Assessment

The methodological quality of each selected study was assessed by two independent reviewers (FG and AML) in accordance with the Quality Assessment of Diagnostic Accuracy Studies 2 (QUADAS-2) criteria [19]. A score of 0.5 points was assigned for an “unclear” assessment, 0 points for “high risk of bias/low applicability” and 1 point for “low risk of bias/high applicability”. Studies with a total of <4 points in the seven QUADAS-2 subdomains were excluded from further analysis.

### 2.4. Data Extraction and Synthesis

Data from the included studies were systematically extracted and consisted of relevant study characteristics, methodologies, and outcomes: authors, year of publication, journal, study design, sample size, thyroid cancer subtype, radiopharmaceutical(s), imaging modality (PET or scintigraphy), time interval between negative WBS and the evaluated imaging study, type of analysis (qualitative or quantitative, per-lesion or per-patient), reference standard, main results, and metrics. When more than one radiopharmaceutical was studied, the results were reported separately. Study characteristics were presented in summary tables. Non-normally distributed continuous variables were reported as median and interquartile range (IQR). Categorical variables were reported as number of studies. Synthesis and analysis of included studies were performed using Excel^®^ 2017 (Microsoft^®^, Redmond, WA, USA).

### 2.5. Meta-Analysis

Only studies focusing on [^18^F]FDG PET/CT with complete data to construct a confusion matrix were eligible for inclusion in this meta-analysis. For each study, the patient-based number of true positive, true negative, false positive, and false negative examinations were extracted. Sensitivity and specificity with their respective 95% confidence intervals (CIs) were determined for each study using a logit transformation with random effects via the “metadta” command in STATA version 17.0 [20]. We applied the fitted random effects model as it proved to be more suitable for the data than a fixed effects model (*p* < 0.0001). Forest plots were generated to display the pooled estimates of sensitivity and specificity, along with their CIs. Between-study heterogeneity was assessed for sensitivity and specificity using generalised tau-squared and I-squared statistics by Zhou and Dendukur [21]. Heterogeneity was categorised as low, moderate, or high, with low/moderate heterogeneity (i.e., I^2^ < 75%) considered acceptable [22]. Statistical analyses were performed using STATA (version 17.0, Stata-Corp LP, College Station, TX, USA). The significance level was set at *p* ≤ 0.05.

## 3. Results

### 3.1. Study Selection and Quality Assessment

The PubMed/MEDLINE and EMBASE searches identified 279 and 259 studies, respectively, resulting in a total of 538 studies. After removing 254 duplicates, 284 articles were assessed for eligibility by title and abstract screening. At this stage, 199 studies were excluded. The remaining 85 studies underwent full-text assessment. The full text of 13 studies was not available, leaving 72 studies. Based on the pre-specified inclusion criteria, 48 studies were not eligible for inclusion. Quality assessment was performed in 24 studies, of which two were excluded for high risk of bias (QUADAS < 4 points). As a result, 22 studies were included in the systematic review. In the meta-analysis phase, studies focusing on [^18^F]FDG PET/CT (*n* = 20) were assessed for inclusion. Complete data necessary to construct a confusion matrix were not available in 3 studies, which were excluded. Finally, 18 studies were included in the meta-analysis. A visual representation of the article selection process is shown in Figure 1.

The main quality issues in the 22 studies included in the review were related to the reference standard and flow and timing domains (Figure 2). The assessment of the quality of the individual studies is shown in Table 1.

### 3.2. Study Characteristics

The systematic review included a comprehensive analysis of 22 studies published between 1999 and 2023 that investigated the diagnostic accuracy of nuclear medicine imaging in detecting disease recurrence in patients with RAI-refractory thyroid cancer. A total of 1299 patients from different studies were included in this review. Half of the included studies (n = 11) had a retrospective design. Except for one study that focused solely on papillary thyroid cancer (PTC) [35], all papers included patients with different thyroid cancer histotypes. The identification of RAI-refractory thyroid cancer was based on post-treatment WBS in eleven studies, while seven studies employed diagnostic WBS. In three trials, both approaches were used, while one trial did not specify its method. The time between negative WBS and subsequent imaging varied between studies, with a median interval of 2 weeks (IQR 1–12). PET/CT and scintigraphy were used as primary imaging modalities in 20/22 and 2/22 studies, respectively. Among PET studies, [^18^F]FDG was the most commonly employed tracer (n = 20). It was compared to [^68^Ga]Ga-DOTA-NOC in two cases, one of which involved [^18^F]FDG-negative patients, and to [^68^Ga]Ga-DOTA-RGD(2) in another case. The two studies using scintigraphy as primary imaging modality explored [^99m^Tc]Tc-EDDA/HYNIC-TOC and [^99m^Tc]Tc-3PRGD2, respectively. Image analysis was qualitative in 14 studies, while seven studies also included semiquantitative data. Diagnostic accuracy was mainly reported at a per-patient level (n = 17). The reference standard was a composite of histopathological findings and clinical-radiological follow-up, except for one study [35] that used histopathology as the reference standard. Table 2 synthesises key features of the included studies.

The main characteristics and results of the included studies are summarised in Table 3 and Table 4, respectively.

### 3.3. Studies Using [^18^F]FDG PET/CT as Primary Imaging Modality

Grünwald et al. [25] showed an increased accuracy of [^18^F]FDG in Hürthle cell carcinoma compared to other histotypes. Palmedo et al. [27] demonstrated high [^18^F]FDG PET/CT diagnostic accuracy in detecting both locoregional and distant metastases (93% and 100%, respectively). Conversely, Miralliè et al. [28] reported moderate levels of sensitivity (63%) and accuracy (53%); however, sensitivity increased in patients with Tg levels above 10 ng/mL. Similarly, Bannas et al. [31] reported an overall accuracy of [^18^F]FDG PET/CT of 66%, which increased to 71% in patients with Tg levels > 10 ng/mL. This increased accuracy led to a change in clinical management in 17 out of 30 patients, guiding curative surgery in 9 cases and altering the initial therapeutic strategy in 8 of them. Kendi et al. [41] showed that positive PET/CT contributed to changes in clinical management in 50% of patients included in their study. Ora et al. [43] reported a detection rate of 65% for [^18^F]FDG PET/CT, with lesions being predominantly localised in the neck and thorax (about 97%). Tg proved to be a significant but moderately predictive factor for PET positivity (AUC 66.5%, *p* = 0.004), whereas TgAb showed no significant correlation (*p* = 0.961). Boktor et al. [44] demonstrated an accuracy of 95% for [^18^F]FDG PET/CT in detecting DTC recurrence, observing a direct correlation with increasing Tg levels. This high diagnostic accuracy significantly impacted patient management in 19 out of 67 patients, leading to redirection towards surgery, systemic therapy, or radiotherapy. Esteva et al. [29] found significant correlations between primary tumour dimensions, thyroid capsular invasion, and [^18^F]FDG positivity. Similarly, Vural et al. [33] identified extrathyroidal spread, tumour size, and Tg levels as independent risk factors associated with [^18^F]FDG-avid recurrence. Overall, they reported a diagnostic accuracy of 75%, underlining that most PET-negative lesions were smaller than 1 cm. Notably, they performed separate analyses to determine the optimal Tg cut-off values in TSH-suppressed and TSH-stimulated patients, resulting in cut-off values of 1.9 ng/mL and 38.2 ng/mL, respectively [33]. Stangierski et al. [39] reported a detection rate per patient of 43%, with negative [^18^F]FDG PET/CT scans occurring predominantly in patients with small lesions below the spatial resolution of PET. Positive patients had significantly higher Tg levels than negative PET/CT patients (143.8 vs. 26.5 ng/mL), establishing an optimal Tg cut-off of 32.9 ng/mL in predicting [^18^F]FDG PET/CT positivity. Consistently with these findings, Kaewput et al. [40] showed that [^18^F]FDG PET/CT had an excellent overall accuracy of 95% and detected previously undiagnosed locoregional lesions and distant metastases in 7 patients. [^18^F]FDG PET sensitivity was 87% in patients with stimulated Tg levels below 50 ng/dL and increased to 100% in those with levels above 50 ng/dL. Giovanella et al. [30] identified a Tg cut-off for [^18^F]FDG PET positivity of 4.6 ng/mL to optimise the timing of imaging, achieving an accuracy of 0.96 in patients with Tg levels above this value. Van Dijk et al. [34] demonstrated the utility of [^18^F]FDG PET/CT in the early work-up following a negative WBS with concurrent detectable Tg, revealing additional tumour localisations in 9 out of 52 patients and leading to changes in clinical management in 7 of these 9 cases. Despite the high proportion of patients with positive PET scans and Tg levels above 2 ng/mL, a clinically useful Tg cut-off value for imaging was not established. Ozkan et al. [35] performed subgroup analyses based on Tg and TgAb levels, showing accuracies of 71% and 80% in the high Tg and TgAb groups, respectively. Optimal Tg cut-off values for [^18^F]FDG positivity were found to be 10.8 ng/mL (92% sensitivity, 28% specificity) and 20.7 ng/mL (75% sensitivity, 55% specificity).

Helal et al. [26] demonstrated the superior efficacy of [^18^F]FDG PET/CT over conventional imaging (bone scan, chest x-ray, neck ultrasonography, chest CT, and/or MRI), identifying previously undetected positive lesions and leading to management changes in 29/37 cases. Notably, the detection rate was higher in advanced stages than in early stages (80% vs. 47%). Hempel et al. [38] investigated the impact of combined [^18^F]FDG PET/CT and cervical MRI in 46 patients, showing accuracies of 89% and 61%, respectively, which were increased to 91% by consensus reading.

In the study of Kunawudhi et al. [32], serial PET scans performed between 10 and 170 min after [^18^F]FDG injection showed that using an SUVmax threshold of 2.75 at 90 min, together with a percentage change in SUVmax between 60 and 90 min of 1.1%, achieved a diagnostic accuracy of identifying malignant lesions of 97%.

### 3.4. Studies Using Other PET Radiopharmaceuticals Alongside [^18^F]FDG PET/CT as Primary Imaging Modality

Kundu et al. [36] performed a comparative analysis between [^68^Ga]Ga-DOTA-TOC and [^18^F]FDG, showing per-patient accuracies of 87% and 82%, respectively. Lesion-based analysis showed that [^18^F]FDG outperformed [^68^Ga]Ga-DOTA-TOC, detecting 168 of 186 lesions compared to 121, with concordance observed in 103 lesions. Binse et al. [37] evaluated the diagnostic accuracy of [^68^Ga]Ga-DOTA-TOC PET/CT in patients found negative on [^18^F]FDG PET/CT. They discovered a detection rate of 33%, stating a higher sensitivity in patients with higher Tg levels and poorly differentiated/oxyphilic carcinomas. In a comparative evaluation of [^68^Ga]Ga-DOTA-RGD2 and [^18^F]FDG PET/CT, Parihar et al. [42] reported similar sensitivity of the two tracers with superior [^68^Ga]Ga-DOTA-RGD2 specificity (100% vs. 50%). Notably, there was a higher incidence of false-positive findings for [^18^F]FDG in neck lymph nodes.

### 3.5. Studies Using Scintigraphy as Primary Imaging Modality

Gabriel et al. [23] evaluated the clinical applicability of a technetium-labelled somatostatin analogue, specifically [^99m^Tc]Tc-EDDA/HYNIC-TOC, in 54 TENIS syndrome patients. They demonstrated excellent specificity despite moderate sensitivity, with superior performance in patients with elevated Tg levels (>30 ng/mL). However, they reported an insufficient spatial resolution of the [^99m^Tc]Tc-labelled radiopharmaceutical in the detection of lesions smaller than 1 cm in diameter. Furthermore, comparing [^99m^Tc]Tc-EDDA/HYNIC-TOC to [^18^F]FDG in a subset of 36 patients, they showed a significant difference in the detection rate of lung metastases in favour of [^18^F]FDG.

Gao et al. [24] evaluated the efficacy of scintigraphy using [^99m^Tc]Tc-3PRGD2, targeting the integrin α(V)β(3) receptor, in the detection of recurrent DTC in 37 patients. The results showed promising diagnostic efficiency, with sensitivity increasing with elevated thyroglobulin (Tg) levels. Notably, sensitivity reached 100% in patients with stimulated Tg levels above 30 ng/mL.

### 3.6. Meta-Analysis

A total of 18 studies focusing on [^18^F]FDG PET/CT were considered eligible for the meta-analysis and collectively included 1023 patients with the TENIS syndrome. All studies provided data on a patient-based analysis. Only five studies provided a separate lesion-based analysis. However, not all data were available for a confusion matrix. Moreover, results could not be stratified by Tg level as not all studies employed the same cut-off values. Therefore, the meta-analysis was limited to a patient-based quantitative synthesis.

The estimated pooled sensitivity and specificity of [^18^F]FDG PET/CT in the detection of RAI-refractory thyroid cancer lesions were 0.87 (95% CI 0.82–0.90) and 0.76 (95% CI 0.61–0.86), respectively (Figure 3). Moderate between-study heterogeneity was observed, with lower heterogeneity in sensitivity (σ2 = 0.26, I^2^ = 42.21%) compared to specificity (σ2 = 1.62, I^2^ = 62.60%). Despite heterogeneity in both dimensions, the bivariate I^2^ was 0.08, and the generalised between-study variance was <0.0001. Figure 4 provides a graphical representation of the Summary Receiver Operating Characteristic (SROC) curve.

## 4. Discussion

Our results confirm the pivotal role of [^18^F]FDG PET/CT in the management of patients with the TENIS syndrome and demonstrate its remarkable diagnostic performance in this particular setting.

The pioneering use of [^18^F]FDG PET for the detection of DTC recurrence dates back to 1987, providing insights into the heterogeneous nature of DTC metastases [45]. Subsequent studies demonstrated increased [^18^F]FDG uptake in metastatic thyroid cancer as a marker of dedifferentiation and poorer prognosis. Dedifferentiation is associated with increased proliferative index and metabolic activity, driven by GLUT1 up-regulation and sodium–iodide symporter down-regulation. The expression level of GLUT1 varies between tumour subtypes, with a spectrum ranging from low expression in DTC to intermediate expression in poorly differentiated thyroid cancer and high expression in anaplastic thyroid cancer [46]. The so-called “flip-flop” phenomenon highlights the inverse correlation between [^18^F]FDG uptake and RAI uptake and tumour differentiation in metastatic thyroid cancer, emphasising the metabolic transition from iodine uptake to increased glucose uptake during tumour cell dedifferentiation [47,48]. The combination of [^18^F]FDG PET/CT and WBS can provide a more comprehensive assessment of tumour burden [49]. Furthermore, it is known that a high [^18^F]FDG uptake is correlated with a reduced benefit from RAI treatment, regardless of WBS findings [50]. Consequently, although [^18^F]FDG PET/CT is not routinary recommended for DTC staging and diagnosis [51], it has emerged as an efficient modality for monitoring recurrent or metastatic DTC, particularly in cases with negative WBS and elevated thyroglobulin levels [3,52,53,54,55], and its use in patients with persistent elevated Tg and negative WBS is recommended by current guidelines [3]. Moreover, [^18^F]FDG PET/CT holds the advantage of molecular imaging being able to perform comparison of metabolic changes in DTC lesions; this can facilitate the evaluation of response in the course of treatment with novel target therapies [56,57] as well as provide prognostic information [58]. Moreover, recent advances in PET/CT instrumentation, particularly the advent of digital tomography, have resulted in improved spatial resolution, thereby increasing lesion detection sensitivity, particularly for lesions smaller than 1 cm [59]. Indeed, compared to an earlier meta-analysis published by Haslerud et al. that summarised data from 1996 to 2014 [47], our study has revealed notable improvements in diagnostic performance with a pooled estimated sensitivity of 0.87 (95% CI 0.82–0.90) vs. 0.76 (95% CI 0.74–0.84), with a trend towards enhanced accuracy in more recent investigations. Accordingly, we would suggest performing “staging” [^18^F]FDG PET/CT in all patients with the TENIS syndrome since it provides both anatomical and functional information. Proper criteria should be used to interpret staging scans, limiting the number of inconclusive reports. Furthermore, the use of recombinant thyroid stimulating hormone (rhTSH) stimulation prior to PET/CT imaging has been shown to improve lesion detection and diagnostic sensitivity [60]. However, in the early phase of the disease, the positive likelihood ratio might be low since [^18^F]FDG PET/CT positivity has shown a positive relationship with the levels of Tg, Tg double time, and anti-Tg antibodies [61,62,63]. We were unable to perform a meta-analysis based on Tg cut-offs due to the lack of detailed results and the considerable heterogeneity in the used cut-off values. Considering the detection limit for [^18^F]FDG PET (on the order of 10^5 to 10^6 cells [64], further improved with new technology) and the amount of Tg secreted by cells (Tg transcripts comprising 2.6% of the entire mRNA pool in human thyrocytes [65]), we could theoretically estimate the minimum Tg value recommendable to maximise the pre-test probability of the scan, supposing the serum Tg linearly correlates with the volume of the disease. However, the production of Tg is influenced by many factors [66,67,68], making the estimation of a reliable number difficult. Several studies included in the present review have attempted to determine optimal cut-off values for Tg levels in [^18^F]FDG PET/CT timing and interpretation, ranging from 10 to 38 ng/mL [28,31,33,35,37,39,40,43,44], although some reports suggest even lower values [30,34]. According to the literature and our experience, in the staging of the TENIS syndrome, we generally suggest the exam when the unstimulated serum Tg value is >5 ng/mL, whenever [^18^F]FDG PET/CT may impact patient management. Once recurrent disease has been identified and treated, [^18^F]FDG PET/CT could be useful to assess treatment response, and a scan with lower values could be justified on the suspicion of residual disease.

Regardless of the cut-off and despite elevated Tg levels, in a subset of patients with the TENIS syndrome, relapses remain undetectable on [^18^F]FDG PET/CT [69]. Alternative radiopharmaceuticals have subsequently been investigated (Table 5). [^68^Ga]-SST analogues PET imaging has emerged as promising for improving metastasis detection [37], potentially paving the way for PRRT [14,70,71,72]. The role of other commonly used radiopharmaceuticals, such as [^11^C]methionine and radiolabelled choline, remains limited and controversial [73,74,75]. Although not covered in this review, PSMA PET/CT imaging has been explored in pilot studies for DTC, including in patients with the TENIS syndrome, and has provided valuable insights [17,76,77,78,79,80,81,82,83]. The underlying biological rationale stems from the increased expression of PSMA within the neo-vasculature of aggressive DTC, opening theranostic strategies exploiting PSMA-targeted radioligand therapy [17,84]. Recently, Moore et al. [85] evaluated the PSMA expression on 91 thyroid cancer specimens, including normal tissue, benign nodules, primary malignant tumours, lymph nodes, and distant metastases, demonstrating that classic PTC, follicular thyroid carcinoma, and RAI-refractory carcinomas presented the highest percentage of PSMA staining positivity among the histological subtypes. Notably, primary tumour PSMA expression and [^18^F]FDG uptake appear to be complementary in prognostic assessment, with positive PSMA expression correlating with increased risk of recurrence and poorer prognosis [86,87]. Similarly, fibroblast activation protein (FAP)-targeted radiopharmaceuticals that bind to tumour-associated fibroblasts, which are increased in dedifferentiated and more aggressive DTCs, have also been investigated [76,88]. Although still in its early stages, experience with FAP-targeted radiopharmaceuticals is promising. FAP-based radiotracers could serve as a viable alternative for imaging and radioligand therapy in patients with the TENIS syndrome or who exhibit low [^18^F]FDG avidity, as therapeutic options for these patients become limited. Some pilot studies on radiolabelled FAP diagnostic and therapy agents have been performed in RAI-refractory patients, with good tumour uptake and promising results, paving the way for further investigations [16,89,90,91,92,93]. Moreover, integrin-targeting radiopharmaceuticals could also represent a mark in RAI-refractory thyroid cancers. Integrin ανβ3, essential for cell migration, invasion, and tumour neoangiogenesis, has been found to be overexpressed in thyroid cancer due to its presence on both activated endothelial cells of the neovasculature and cancer cells’ surface [76]. Arg-Gly-Asp (RGD)-based radiotracers that bind to integrin ανβ3 have been investigated in our review, with encouraging diagnostic performances [24,42].

Various scintigraphic radiopharmaceuticals, including [^99m^Tc]Tc-EDDA/HYNIC-TOC and [^99m^Tc]Tc-3PRGD2, have also been investigated in this clinical context [23,24]. However, the widespread availability of PET/CT and the lower resolution of SPECT images, which hampers visualisation of small lesions, limit their usefulness on a larger scale.

## 5. Conclusions

This review highlights the great potential of nuclear medicine in guiding therapeutic strategies for RAI-refractory thyroid cancer. [^18^F]FDG PET/CT remains central to the detection of disease recurrence in patients with the TENIS syndrome. The emergence of novel radiopharmaceuticals with specific molecular targets is a promising way to overcome the limitations of [^18^F]FDG in these patients—potentially dissolving the mismatch between Tg levels and imaging findings towards the match point of theranostics.

## Figures and Tables

**Figure 1 jcm-13-05362-f001:**
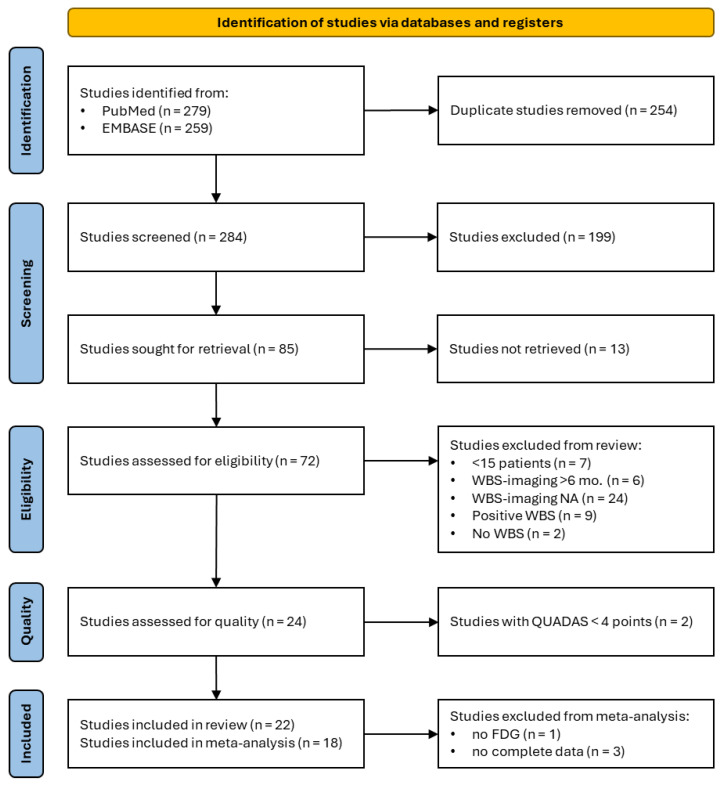
Flow diagram of the article selection process.

**Figure 2 jcm-13-05362-f002:**
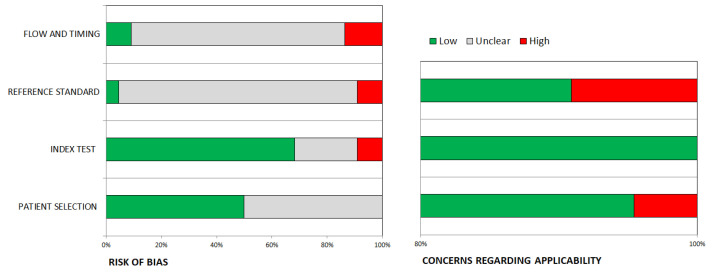
QUADAS-2 quality assessment of the 22 articles included in the systematic review.

**Figure 3 jcm-13-05362-f003:**
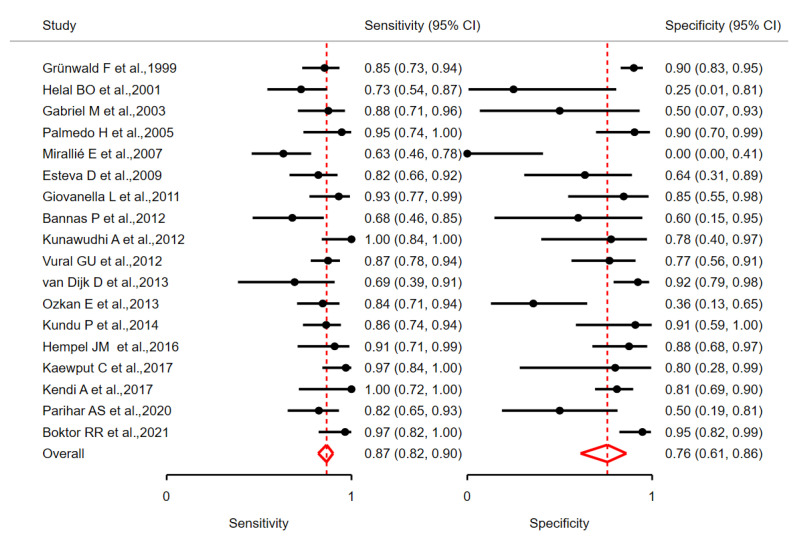
Estimated per-patient pooled sensitivity and specificity of [^18^F]FDG PET/CT in the detection of dedifferentiated thyroid cancer lesions in patients with TENIS syndrome, in studies included in the meta-analysis [23,25,26,27,28,29,30,31,32,33,34,35,36,38,40,41,42,44].

**Figure 4 jcm-13-05362-f004:**
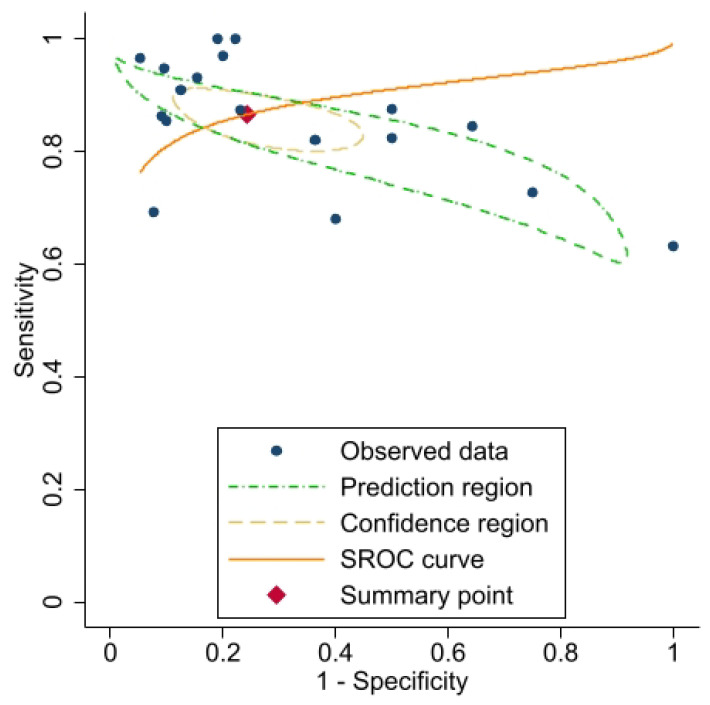
SROC curve of the performance of [^18^F]FDG PET/CT in the detection of dedifferentiated thyroid cancer lesions in patients with TENIS syndrome.

**Table 1 jcm-13-05362-t001:** Risk of bias assessment in each individual study included in the systematic review.

Study	Risk of Bias				Applicability Concerns		
	Patient Selection	Index Test	Reference Standard	Flow and Timing	Patient Selection	Index Test	Reference Standard
Gabriel, M. et al., 2003 [23]	?	☺	?	?	☹	☺	☺
Gao, R. et al. [24]	☺	☺	?	?	☺	☺	☺
Grünwald, F. et al. [25]	?	?	?	?	☺	☺	☺
Helal, B.O. et al. [26]	☺	☺	☹	☹	☺	☺	☺
Palmedo, H. et al. [27]	?	☺	?	?	☺	☺	☺
Mirallié, E. et al. [28]	?	☺	?	?	☺	☺	☺
Esteva, D. et al. [29]	?	☺	?	?	☺	☺	☺
Giovanella, L. et al. [30]	☺	?	?	?	☺	☺	☺
Bannas, P. et al. [31]	☺	☺	?	?	☺	☺	☺
Kunawudhi, A. et al. [32]	☺	☺	?	☹	☺	☺	☺
Vural, G.U. et al. [33]	?	?	?	☺	☺	☺	☺
van Dijk, D. et al. [34]	?	☺	?	?	☺	☺	☺
Ozkan, E. et al. [35]	?	☺	?	☺	☺	☺	☺
Kundu, P. et al. [36]	☺	☺	?	?	☺	☺	☺
Binse, I. et al. [37]	☺	☺	☹	☹	☺	☺	☺
Hempel, J.M. et al. [38]	☺	☺	☺	?	☺	☺	☺
Stangierski, A. et al. [39]	?	?	?	?	☺	☺	☹
Kaewput, C. et al. [40]	?	☹	?	?	☺	☺	☺
Kendi, A. et al. [41]	☺	?	?	?	☺	☺	☹
Parihar, A.S. et al. [42]	☺	☺	?	?	☺	☺	☺
Ora, M. et al. [43]	?	☹	?	?	☺	☺	☺
Boktor, R.R. et al. [44]	☺	☺	?	?	☺	☺	☺

☺ Low Risk; ☹ High Risk; ? Unclear Risk.

**Table 2 jcm-13-05362-t002:** Overall summary of included study characteristics.

Study Characteristics		Included Studies (n = 22)
Patients	n < 50	12
	n ≥ 50	10
Study design	Retrospective	11
	Prospective	9
	N/A	2
Thyroid cancer histotype	Papillary	1
	Mixed	20
WBS	Post-treatment	11
	Diagnostic	7
	Mixed	3
	N/A	1
Imaging modality	PET/CT	20
	Scintigraphy	2
Diagnostic accuracy analysis	Per-patient	17
	Per-patient and per-lesion	5

**Table 3 jcm-13-05362-t003:** Main characteristics of studies included in the systematic review.

Study	Year	Design	No. Patients	TC Subtype	Imaging Modality	RPh	WBS Type	WBS-Imaging Timing (Weeks)	Reference Standard	Image Analysis
Gabriel, M. et al. [23]	2003	P	54	PTC, FTC, OC	Scintigraphy SPECT/CT	[^99m^Tc]Tc-EDDA/HYNIC-TOC	T	12	histopathological results, other imaging methods (US, CT, MRI)	Qualitative
			36/54		PET/CT	[^18^F]FDG				Qualitative
Gao, R. et al. [24]	2018	P	37	PTC, FTC	SPECT/CT	[^99m^Tc]Tc-3PRGD2	D	1	histopathological results, serial radiological or clinical follow-up	Qualitative and semiquantitative
Grünwald, F. et al. [25]	1999	R	166	PTC, FTC, HCC	PET/CT	[^18^F]FDG	T	4	histopathological results, other imaging methods (US, CT), thyroglobulin level, clinical follow-up	Qualitative
Helal, B.O. et al. [26]	2001	P	37	PTC, FTC, HCC	PET/CT	[^18^F]FDG	T	12	histopathological results, other imaging methods (US, CT, MRI), clinical follow-up	Qualitative
Palmedo, H. et al. [27]	2005	P	40	PTC, FTC, HCC	PET/CT	[^18^F]FDG	T, D	1	histopathological results or clinical follow-up	Qualitative
Mirallié, E. et al. [28]	2007	P	45	PTC, FTC, HCC	PET/CT	[^18^F]FDG	T	12	histopathological results, other imaging methods (US, CT), postoperative Tg levels	Qualitative
Esteva, D. et al. [29]	2009	R	50	PTC, FTC, HCC	PET/CT	[^18^F]FDG	D	1	histopathological results, other imaging methods (US, CT, MRI), 12-month clinical follow-up	Qualitative
Giovanella, L. et al. [30]	2011	N/A	42	PTC, FTC	PET/CT	[^18^F]FDG	T	18	histopathological results, other imaging methods (US, CT, MRI), clinical follow-up	Qualitative
Bannas, P. et al. [31]	2012	R	30	PTC, FTC	PET/CT	[^18^F]FDG	D	0.5	Initial Tg levels, histopathological results, other imaging methods (US, CT, MRI), clinical follow-up	Qualitative
Kunawudhi, A. et al. [32],	2012	P	30	PTC, FTC	PET/CT	[^18^F]FDG	T	24	histopathological results, imaging follow-up (US, CT, [^99m^Tc]Tc-MIBI SPECT/CT, or follow-up FDG PET/CT), Tg levels	Qualitative and quantitative
Vural, G.U. et al. [33]	2012	P	105	DTC (histotypes NA)	PET/CT	[^18^F]FDG	T	1	histopathological results or clinical follow-up	Qualitative
van Dijk, D. et al. [34]	2013	R	52	PTC, FTC, HCC	PET/CT	[^18^F]FDG	T	12	Tg levels, histopathological results, imaging follow-up (CT, MRI, US), follow-up 131I-WBS	Qualitative
Ozkan, E. et al. [35]	2013	N/A	59	PTC	PET/CT	[^18^F]FDG	D	12	histopathological results	Qualitative and quantitative
Kundu, P. et al. [36]	2014	P	62	PTC, FTC	PET/CT	[^68^Ga]Ga-DOTANOC	T, D	2	histopathological results, serial follow-up with serum Tg, clinical examination, response to redifferentiating drugs, and conventional imaging (CT/MRI)	Qualitative and quantitative
					PET/CT	[^18^F]FDG				Qualitative and quantitative
Binse, I. et al. [37]	2016	R	15	PTC, FTC, OTC, poorly differentiated carcinoma	PET/CT	[^68^Ga]Ga-DOTATOC	D	15	histopathological results or clinical follow-up, including different imaging modalities	Qualitative and quantitative
Hempel, J.M. et al. [38]	2016	R	46	PTC, FTC, poorly differentiated carcinoma, anaplastic carcinoma	PET/CT	[^18^F]FDG	T	0.5	histopathological results, long-term follow-up (minimum 3 years)	Qualitative
					MRI					Qualitative
Stangierski, A. et al. [39]	2016	R	69	DTC (histotypes N/A)	PET/CT	[^18^F]FDG	T	2	histopathological results or clinical follow-up	Qualitative
Kaewput, C. et al. [40]	2017	R	38	PTC, FTC	PET/CT	[^18^F]FDG	T, D	24	histopathological results, other imaging methods (US, CT, MRI), follow-up 131I-WBS, subsequent Tg and TgAb levels	Qualitative and quantitative
Kendi, A. et al. [41]	2017	R	74	DTC (histotypes N/A)	PET/CT	[^18^F]FDG	D	0 (same session)	unclear (histopathology, 131I-WBS, Tg and TgAb levels, US)	Qualitative
Parihar, A.S. et al. [42]	2019	P	44	PTC, FTC	PET/CT	[^18^F]FDG	T	1	histopathological results, other imaging methods (US, CT, MRI), clinical examination, TSH levels, Tg levels, TgAb levels, response to treatment with redifferentiation agents/tyrosine kinase inhibitors, local radiation therapy	Qualitative and quantitative
					PETCT	[^68^Ga]Ga-DOTA-RGD_2_				Qualitative and quantitative
Ora, M. et al. [43]	2020	R	137	PTC, FTC	PET/CT	[^18^F]FDG	D	1	histopathological results, CT imaging, persistently raised Tg or TgAb levels in follow-up	Qualitative
Boktor, R.R. et al. [44]	2021	R	67	PTC, FTC, HCC, insular carcinoma, poorly differentiated carcinoma	PET/CT	[^18^F]FDG	N/A	2	histopathological results, imaging follow-up, clinical follow-up, Tg levels	Qualitative

Abbreviations: TC—thyroid cancer; WBS—whole-body scan; P—prospective; R—retrospective; PTC—papillary thyroid carcinoma; FTC—follicular thyroid carcinoma; HCC—Hürthle-cell carcinoma; SPECT/CT—single photon emission tomography/computed tomography; PET/CT—positron emission tomography/computed tomography; T—therapy; D—diagnostic; MRI—magnetic resonance imaging; US—ultrasound; CT—computed tomography; Tg—thyroglobulin; TgAb—thyroglobulin antibodies; N/A—not applicable.

**Table 4 jcm-13-05362-t004:** Diagnostic performance of studies included in the systematic review.

Study	Imaging Modality	Radiopharmaceutical	Type of Analysis	Sensitivity (%)	Specificity (%)	Accuracy (%)
Gabriel, M. et al. [23]	Scintigraphy SPECT/CT	[^99m^Tc]Tc-EDDA/HYNIC-TOC	Per patient	66	100	68.5
	PET/CT	[^18^F]FDG		87.5	50	83.3
Gao, R. et al. [24]	SPECT/CT	[^99m^Tc]Tc-3PRGD2	Per patient	96.6	75	NA
Grünwald, F. et al. [25]	PET/CT	[^18^F]FDG	Per patient	85	90	89
Helal, B.O. et al. [26]	PET/CT	[^18^F]FDG	Per patient	N/A	N/A	N/A
Palmedo, H. et al. [27]	PET/CT	[^18^F]FDG	Per patient and per lesion	95	91	93
Mirallié, E. et al. [28]	PET/CT	[^18^F]FDG	Per patient	63	N/A	53
Esteva, D. et al. [29]	PET/CT	[^18^F]FDG	Per patient	82	64	N/A
Giovanella, L. et al. [30]	PET/CT	[^18^F]FDG	Per patient	93	84	90
Bannas, P. et al. [31]	PET/CT	[^18^F]FDG	Per patient	68	60	66.7
Kunawudhi, A. et al. [32]	PET/CT	[^18^F]FDG	Per patient and per lesion	100	78	93
Vural, G.U. et al. [33]	PET/CT	[^18^F]FDG	Per patient	87	77	75
van Dijk, D. et al. [34]	PET/CT	[^18^F]FDG	Per patient	69	92	N/A
Ozkan, E. et al. [35]	PET/CT	[^18^F]FDG	Per patient and per lesion	82	30	71
Kundu, P. et al. [36]	PET/CT	[^68^Ga]Ga-DOTANOC	Per patient and per lesion	78.4	100	82.3
	PET/CT	[^18^F]FDG		86.3	90.9	87
Binse, I. et al. [37]	PET/CT	[^68^Ga]Ga-DOTATOC	Per patient	N/A	N/A	N/A
Hempel, J.M. et al. [38]	PET/CT	[^18^F]FDG	Per patient	91	87	89
	MRI			54	67	61
Stangierski, A. et al. [39]	PET/CT	[^18^F]FDG	Per patient	N/A	N/A	N/A
Kaewput, C. et al. [40]	PET/CT	[^18^F]FDG	Per patient	96.9	80	94.7
Kendi, A. et al. [41]	PET/CT	[^18^F]FDG	Per patient	N/A	N/A	N/A
Parihar, A.S. et al. [42]	PET/CT	[^18^F]FDG	Per patient and per lesion	82	50	75
	PET/CT	[^68^Ga]Ga-DOTA-RGD_2_		82.3	100	86.4
Ora, M. et al. [43]	PET/CT	[^18^F]FDG	Per patient	N/A	N/A	N/A
Boktor, R.R. et al. [44]	PET/CT	[^18^F]FDG	Per patient	96.5	94.5	95.5

Abbreviations: SPECT/CT—single photon emission tomography/computed tomography; PET/CT—positron emission tomography/computed tomography; MRI—magnetic resonance imaging; N/A—not applicable.

**Table 5 jcm-13-05362-t005:** Summary of PET tracers compared to [^18^F]FDG in TENIS syndrome.

Tracer	Main Result	Future Perspectives *
Choline	<[^18^F]FDG	±
Methionine	<[^18^F]FDG	-
Prostate specific membrane antigen	>[^18^F]FDG	+ (++ in RLT selection)
Integrin ανβ3	∽[^18^F]FDG	±
Fibroblast Activation Protein	>[^18^F]FDG	+ (++ in RLT selection)
Somatostatin analogues	<[^18^F]FDG	± (++ in PRRT selection)

++ suitable; + promising; ± uncertain; - non indicated; * based on our judgement.

## Data Availability

The data presented in this systematic review are available in referenced articles.

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
