# Peer review of "Match Point: Nuclear Medicine Imaging for Recurrent Thyroid Cancer in TENIS Syndrome—Systematic Review and Meta-Analysis"

_jcm, 2024, doi:10.3390/jcm13185362_

Round 1
Reviewer 1 Report
Comments and Suggestions for Authors
For the authors:
The author brings a systematic review and meta-analysis that comprehensively evaluates the diagnostic capabilities of nuclear imaging in TENIS syndrome for RAIR-DTC patients. It provided pooled estimates of sensitivity and specificity for [18F] FDG PET/CT, and also discusses the potential of emerging radiopharmaceuticals to improve diagnostic accuracy and therapeutic approaches. This manuscript is well-crafted, yet there are several questions that require resolution before its consideration for acceptance.
1. 131I-WBS+SPECT/CT is a pivotal diagnostic tool during the 131I treatment of DTC. It serves multiple critical functions, including initial assessment, efficacy evaluation, lesion detection, restaging, treatment guidance, and prognosis determination. A well-established consensus among experts is already in place to guide clinical management. (ref.1).
2. The study primarily focuses on the value of nuclear medicine molecular imaging in RAIR-DTC. It is observed that as the degree of differentiation in DTC lesions decreases, their invasiveness increases, often exhibiting a 'flip-flop' phenomenon where the iodine uptake capability and the affinity for 18F-FDG are inversely related. Regardless of iodine uptake by the lesions, concurrent increased 18F-FDG uptake suggests a reduced benefit from 131I therapy (ref. 2). The combination of 18F-FDG PET/CT with 131I-WBS+SPECT/CT allows for a comprehensive assessment of tumor burden (ref. 3). The authors are encouraged to delineate how their research advances upon these existing studies. Furthermore, the authors calculated pooled sensitivity and specificity using a random-effects model, yielding a sensitivity of 0.87 (95% Confidence Interval: 0.82-0.90) and a specificity of 0.76 (95% Confidence Interval: 0.61-0.86). It is pertinent to discuss whether these sensitivity and specificity figures are more suitable for 'rolling in' or 'rolling out' patients, and to determine if they are more appropriate for identifying potential patients at the time of diagnosis or for monitoring recurrence in post-surgical patients.
3. Molecular imaging holds a distinct advantage in the early assessment of therapeutic efficacy for RAIR-DTC. The use of 18F-FDG PET/CT, in conjunction with the standardized criteria for PERCIST (PET response criteria in solid tumors), enables a quantitative comparison of the metabolic changes in RAIR-DTC lesions. This method is instrumental in evaluating the metabolic response to targeted drug therapies (ref.4-5) and has been correlated with Progression-Free Survival (PFS) (ref.6). It is imperative to provide a thorough discussion on the monitoring and management of adverse effects associated with RAIR-DTC treatments.
4. In addition to the established 18F-FDG PET/CT, emerging radiotracers (ref.7), fibroblast activation protein inhibitors (FAPI) labeled with radionuclides (ref.8) and somatostatin receptor analogs (ref.9-10), are increasingly being employed for the detection of RAIR-DTC. Given the potential of these novel agents, it is suggested that the authors incorporate a discussion on these considerations, exploring their implications and potential advantages over traditional imaging modalities.
reference:
1. Nuclear Medicine Expert Committee of Chinese Society of Clinical Oncology, Thyroid Cancer Expert Committee of Chinese Society of Clinical Oncology, Chinese Society of Nuclear Medicine, et al. Management guidelines for radioactive iodine-refractory differentiated thyroid cancer (2024 edition) [J]. Chin J Nucl Med Mol Imaging, 2024,44(06):359-372. DOI:10.3760/cma.j.cn321828-20240125-00034
2. Robbins RJ, Wan Q, Grewal RK, Reibke R, Gonen M, Strauss HW, Tuttle RM, Drucker W, Larson SM. Real-time prognosis for metastatic thyroid carcinoma based on 2-[18F]fluoro-2-deoxy-D-glucose-positron emission tomography scanning. J Clin Endocrinol Metab. 2006 Feb;91(2):498-505. doi: 10.1210/jc.2005-1534. Epub 2005 Nov 22. PMID: 16303836.
3. Albano D, Dondi F, Mazzoletti A, Bellini P, Rodella C, Bertagna F. Prognostic Role of 2-[18F]FDG PET/CT Metabolic Volume Parameters in Patients Affected by Differentiated Thyroid Carcinoma with High Thyroglobulin Level, Negative 131I WBS and Positive 2-[18F]-FDG PET/CT. Diagnostics (Basel). 2021 Nov 25;11(12):2189. doi: 10.3390/diagnostics11122189. PMID: 34943426; PMCID: PMC8700137.
4. Ferrari C, Santo G, Ruta R, Lavelli V, Rubini D, Mammucci P, Sardaro A, Rubini G. Early Predictive Response to Multi-Tyrosine Kinase Inhibitors in Advanced Refractory Radioactive-Iodine Differentiated Thyroid Cancer: A New Challenge for [18F]FDG PET/CT. Diagnostics (Basel). 2021 Aug 5;11(8):1417. doi: 10.3390/diagnostics11081417. PMID: 34441351; PMCID: PMC8392185.
5. Valerio L, Guidoccio F, Giani C, Tardelli E, Puccini G, Puleo L, Minaldi E, Boni G, Elisei R, Volterrani D. [18F]-FDG-PET/CT Correlates With the Response of Radiorefractory Thyroid Cancer to Lenvatinib and Patient Survival. J Clin Endocrinol Metab. 2021 Jul 13;106(8):2355-2366. doi: 10.1210/clinem/dgab278. PMID: 33901285.
6. Ahmaddy F, Burgard C, Beyer L, Koehler VF, Bartenstein P, Fabritius MP, Geyer T, Wenter V, Ilhan H, Spitzweg C, Todica A. 18F-FDG-PET/CT in Patients with Advanced, Radioiodine Refractory Thyroid Cancer Treated with Lenvatinib. Cancers (Basel). 2021 Jan 16;13(2):317. doi: 10.3390/cancers13020317. PMID: 33467085; PMCID: PMC7830971.
7. de Vries LH, Lodewijk L, Braat AJAT, Krijger GC, Valk GD, Lam MGEH, Borel Rinkes IHM, Vriens MR, de Keizer B. 68Ga-PSMA PET/CT in radioactive iodine-refractory differentiated thyroid cancer and first treatment results with 177Lu-PSMA-617. EJNMMI Res. 2020 Mar 6;10(1):18. doi: 10.1186/s13550-020-0610-x. PMID: 32144510; PMCID: PMC7060303.
8. Fu H, Wu J, Huang J, Sun L, Wu H, Guo W, Qiu S, Chen H. 68Ga Fibroblast Activation Protein Inhibitor PET/CT in the Detection of Metastatic Thyroid Cancer: Comparison with 18F-FDG PET/CT. Radiology. 2022 Aug;304(2):397-405. doi: 10.1148/radiol.212430. Epub 2022 May 10. PMID: 35536131.
9. Kundu P, Lata S, Sharma P, Singh H, Malhotra A, Bal C. Prospective evaluation of (68)Ga-DOTANOC PET-CT in differentiated thyroid cancer patients with raised thyroglobulin and negative (131)I-whole body scan: comparison with (18)F-FDG PET-CT. Eur J Nucl Med Mol Imaging. 2014 Jul;41(7):1354-62. doi: 10.1007/s00259-014-2723-9. Epub 2014 Feb 22. PMID: 24562651.
10. Basu S, Kalshetty A, Fargose P. Interlesional 'flip-flop' between 68Ga-DOTATATE and FDG-PET/CT in thyroglobulin-elevated negative iodine scintigraphy (TENIS) syndrome. Natl Med J India. 2017 Jan-Feb;30(1):48. PMID: 28731009.
Author Response
We thank the reviewer for the valuable comments and suggestions.
Comment 1: "131I-WBS+SPECT/CT is a pivotal diagnostic tool during the 131I treatment of DTC. It serves multiple critical functions, including initial assessment, efficacy evaluation, lesion detection, restaging, treatment guidance, and prognosis determination. A well-established consensus among experts is already in place to guide clinical management. (ref.1)"
Response: We expanded the Introduction to better delineate the role of WBS, according to Reviewer's suggestion (lines 42-44).
Comment 2: "The study primarily focuses on the value of nuclear medicine molecular imaging in RAIR-DTC. It is observed that as the degree of differentiation in DTC lesions decreases, their invasiveness increases, often exhibiting a 'flip-flop' phenomenon where the iodine uptake capability and the affinity for 18F-FDG are inversely related. Regardless of iodine uptake by the lesions, concurrent increased 18F-FDG uptake suggests a reduced benefit from 131I therapy (ref. 2). The combination of 18F-FDG PET/CT with 131I-WBS+SPECT/CT allows for a comprehensive assessment of tumor burden (ref. 3). The authors are encouraged to delineate how their research advances upon these existing studies. Furthermore, the authors calculated pooled sensitivity and specificity using a random-effects model, yielding a sensitivity of 0.87 (95% Confidence Interval: 0.82-0.90) and a specificity of 0.76 (95% Confidence Interval: 0.61-0.86). It is pertinent to discuss whether these sensitivity and specificity figures are more suitable for 'rolling in' or 'rolling out' patients, and to determine if they are more appropriate for identifying potential patients at the time of diagnosis or for monitoring recurrence in post-surgical patients."
Response: We expanded the Discussion enriching it with further considerations on the role of [18F]FDG, and on its precise potential role in patient enrolment and management, according to Reviewer's suggestions (lines 319-343).
Comment 3: "Molecular imaging holds a distinct advantage in the early assessment of therapeutic efficacy for RAIR-DTC. The use of 18F-FDG PET/CT, in conjunction with the standardized criteria for PERCIST (PET response criteria in solid tumors), enables a quantitative comparison of the metabolic changes in RAIR-DTC lesions. This method is instrumental in evaluating the metabolic response to targeted drug therapies (ref.4-5) and has been correlated with Progression-Free Survival (PFS) (ref.6). It is imperative to provide a thorough discussion on the monitoring and management of adverse effects associated with RAIR-DTC treatments."
Response: We expanded the Discussion to touch on the role of [18]FDG in the management of patients undergoing target therapies, according to Reviewer's suggestions (lines 302-305 and 309-312).
Comment 4: "In addition to the established 18F-FDG PET/CT, emerging radiotracers (ref.7), fibroblast activation protein inhibitors (FAPI) labeled with radionuclides (ref.8) and somatostatin receptor analogs (ref.9-10), are increasingly being employed for the detection of RAIR-DTC. Given the potential of these novel agents, it is suggested that the authors incorporate a discussion on these considerations, exploring their implications and potential advantages over traditional imaging modalities."
Response: We enriched our Discussion in the paragraphs dedicated to novel radiopharmaceuticals according to Reviewer's suggestions (lines 344-380).
Reviewer 2 Report
Comments and Suggestions for Authors Dear Editor and Authors,Thank you for the opportunity to review this manuscript.
The work is of interest and provides a valuable meta-analysis summarizing the evidence on nuclear medicine in RAI-refractory thyroid cancer.
The manuscript is clearly written, with adherence to PRISMA guidelines, and the meta-analysis was conducted properly. The results are well presented and discussed.
I have only two minor comments:
1. The decision regarding whether to perform a fixed-effects or random-effects model should be stated in the Methods section. 2. In Figure 2, the panel on "Concerns regarding applicability" appears to differ from the assessment presented in Table 1.
Thank you once again.
Best regards,
Author Response
We thank the reviewer for the valuable comments and suggestions.
Comment 1: "The decision regarding whether to perform a fixed-effects or random-effects model should be stated in the Methods section."
Response: We moved the sentence explaining the chosen model from the Results to the Methods section (lines 120-122).
Comment 2: "In Figure 2, the panel on "Concerns regarding applicability" appears to differ from the assessment presented in Table 1."
Response: We updated Figure 2 accordingly.
Round 2
Reviewer 1 Report
Comments and Suggestions for Authors
The author has made significant revisions to the manuscript, an effort that deserves commendation. However, to ensure the coherence and consistency of the manuscript's search strategy, especially within the professional domain, it is recommended that the author pays special attention to updating the search strategy to reflect the revised content, including but not limited to:
l Figure 1. Flow diagram of the article selection process.
l Section 3.2. Study characteristics
l Table 2. Overall summary of included study characteristics.
Additionally, all revised references should be consistent with the search strategy to ensure the transparency and replicability of the research. Moreover, the author should strictly adhere to the PRISMA guidelines to ensure the scientific rigor and standardized reporting of the research methods. This meticulous adjustment is crucial for maintaining the overall coherence of the manuscript's search strategy and enhances the persuasiveness of the manuscript.
Author Response
Comment 1:
The author has made significant revisions to the manuscript, an effort that deserves commendation. However, to ensure the coherence and consistency of the manuscript's search strategy, especially within the professional domain, it is recommended that the author pays special attention to updating the search strategy to reflect the revised content, including but not limited to:
l Figure 1. Flow diagram of the article selection process.
l Section 3.2. Study characteristics
l Table 2. Overall summary of included study characteristics.
Additionally, all revised references should be consistent with the search strategy to ensure the transparency and replicability of the research. Moreover, the author should strictly adhere to the PRISMA guidelines to ensure the scientific rigor and standardized reporting of the research methods. This meticulous adjustment is crucial for maintaining the overall coherence of the manuscript's search strategy and enhances the persuasiveness of the manuscript.
Response 1:
We thank the reviewer for the valuable comments and suggestions. In the revised version of the review, as suggested by Reviewers, we added references in the Introduction and Discussion sections and the bibliography has been updated accordingly. We did not modify the Materials and Methods or the Results sections; specifically, we did not perform substantial changes the research strategy, study selection, or analysis.
However, by double checking Tables 1 e 2 we found a wrongly linked reference number, caused by a trasposition error during reference update in the review process. We thank the Reviewer for pointing this out. We updated the references accordingly.
This systematic review was conducted in accordance with the PRISMA guidelines.